

# Use of necrophagous insects as evidence of cadaver relocation: myth or reality?

Damien Charabidze[1], Matthias Gosselin[2] and Valéry Hedouin[1]

[1] CHU Lille, EA 7367 UTML – Unite de Taphonomie Medico-Legale, Univ Lille, Lille, France
[2] Research Institute of Biosciences, Laboratory of Zoology, UMONS – Université de Mons, Mons, Belgium

## ABSTRACT

The use of insects as indicators of post-mortem displacement is discussed in many texts, courses and TV shows, and several studies addressing this issue have been published. Although the concept is widely cited, it is poorly understood, and only a few forensic cases have successfully applied such a method. The use of necrophagous insects as evidence of cadaver relocation actually involves a wide range of biological aspects. Distribution, microhabitat, phenology, behavioral ecology, and molecular analysis are among the research areas associated with this topic. This article provides the first review of the current knowledge and addresses the potential and limitations of different methods to evaluate their applicability. This work reveals numerous weaknesses and erroneous beliefs as well as many possibilities and research opportunities.

# INTRODUCTION

## Context

This article reviews the current knowledge and potential of different methods involving the use of insects to study cadaver relocation and evaluates their feasibilities.

Insect analysis has been used in legal investigations for decades in a practice now known as forensic entomology (*Benecke, 2001*). Increased interest in this field since the late 20th century has resulted in more frequent use of forensic entomology in investigations and the development of research on necrophagous species. Previous reviews have compiled and explained the aims and methods of forensic entomology (*Catts & Goff, 1992*; *Byrd & Castner, 2009*, *Bala, 2015*), but some fundamental questions remain unresolved, particularly the potential use of insects as evidence of corpse relocation.

Cadaver decomposition can be altered due to human activity, especially steps taken to hide a cadaver (*Haglund & Sorg, 1997*; *Mann, Bass & Meadows, 1990*). Attempts to prevent discovery include cadaver concealment, wrapping, and displacement. Such post-mortem relocation can occur shortly after death or after days of concealment. It can occur over short (e.g., from the room where death occurred to the garden of the house) or long distances. In most cases, the location where the cadaver is hidden is very different from that where death occurred (*Reibe et al., 2008*). Currently, only a few scientific methods are

Corresponding author
Damien Charabidze,
damien.charabidze@univ-lille2.fr

available to determine whether a crime scene is a primary or secondary scene (*Miller Coyle et al., 2005*). Because the biology and ecology of necrophagous species can convey information on where and how insects live, forensic manuals and courses often state that insects can be used as evidence of cadaver relocation (*Catts & Goff, 1992*; *Byrd & Castner, 2009*; *Smith, 1986*; *Joseph et al., 2011*; *Mozayani & Noziglia, 2011*; *Archer & Wallman, 2016*; *Payne-James & Byard, 2015*). However, while this idea is appealing, it may not reflect reality.

It may seem obvious that "if a body is discovered with insects restricted to a habitat or geographic region different from that in which it is discovered, this is an indication that the body may have been moved following death" (*Mozayani & Noziglia, 2011*). However, most, if not all, European necrophagous species have large distribution areas covering many countries and hundreds of thousands of square kilometers, making the sampling of non-native species within European regions unlikely. Furthermore, while each species has an ecological niche (e.g., forest or synanthropic environment; sunny or shady habitats), individuals can sometimes occur outside the preferred range. Additionally, the long dispersal capability of most necrophagous species, notably blow flies, makes it difficult to relate a given species to a particular place or habitat and thus draw inferences regarding cadaver relocation (*Nazni et al., 2005*; *Bomphrey, Walker & Taylor, 2009*, *Zabala, Díaz & Saloña-Bordas, 2014*).

Temporal separation is another characteristic of necrophagous species. The phenology (cyclic and seasonal phenomena) of blow flies is well known; some species are primarily active during hot weather, whereas others are adapted to cold climates (*Voss, Spafford & Dadour, 2009*). Under some circumstances, such seasonality might contribute useful information regarding the chronology of cadaver decomposition. However, the presence of larvae of a summer species on a cadaver discovered in the winter does not constitute indisputable evidence of cadaver relocation. In addition, colonization time is strongly dependent on the stage of decomposition. Although it is far more complex than squads (*Wyss, Cherix & Mangin, 2013*), succession on cadavers has been experimentally shown in several countries and under a variety of conditions (*Anderson & Van Laerhoven, 1996*; *Abd El-Bar & Sawaby, 2011*; *Abouzied, 2014*; *Archer, 2014*). Divergence from known succession patterns, such as the absence of certain species or unusual associations, might indicate cadaver relocation or concealment. The presence or absence of some instars is also of relevance, especially with regard to wandering larvae or pupae of pioneer species (e.g., Calliphoridae flies), which pupate away from the cadaver and can thus be found after cadaver removal.

Advances in genetics also offer numerous opportunities. Genetic analyses allow the assignment of individuals to local populations or even sub-populations. As noted by *Tomberlin et al. (2011)*, such analyses are of great interest in the context of cadaver relocation. In addition, the genetic analysis of gut contents has potential for identifying the cadaver that larvae have been feeding on (*Campobasso et al. (2005)* and *Calvignac-Spencer et al. (2013)*). This technique should be developed in the coming years and provide new tools for forensic entomologists and crime scene investigations.
This review analyzes in detail these different approaches. It reveals weaknesses and mistaken beliefs regarding the use of forensic entomology as evidence of cadaver displacement as well as many promising aspects and development opportunities.

## Survey methodology

The first phase of the survey involved identifying the magnitude of the issue of cadaver relocation. This phase was performed by searching the main forensic literature published in English over the last 40 years for studies addressing corpse relocation. We found references to this idea in several forensic entomology books and studies (*Catts & Goff, 1992*; *Byrd & Castner, 2009*; *Smith, 1986*; *Joseph et al., 2011*; *Mozayani & Noziglia, 2011*; *Archer & Wallman, 2016*; *Payne-James & Byard, 2015*), but few case reports (*Goff, 1991*; *Benecke, 1998*; *Krikken & Huijbregts, 2001*). We also found several research articles addressing corpse relocation as a main goal or claiming the potential application of their findings to this topic. From this dataset, we outlined various facets of the problem and gathered them into four main concepts: spatial separation, behavior/development, phenology/colonization time, and molecular analyses. Then, we searched the literature within each of these fields for data on the potential use of different methods to study corpse relocation. The resulting dataset was then analyzed to identify discrepancies or methods with strong application potential.

Due to the number of factors that can affect insect occurrence on corpses (e.g., species, climates, and geography), we focus on the central European area. However, most of the conclusions of this review can be generalized to other locations.

## SPATIAL SEPARATION

Only a few hundred insect species are associated with cadavers, of which a few dozen are strictly necrophagous (requiring a cadaver to feed on during at least a part of their development) (*Smith, 1986*). Their diversity is apparent from their variability in size, shape, behavior, ecological niche, and distribution and reflects species-specific adaptations, which allow species to exploit different habitats and resources. Johnson defined four orders of habitat selection, ranging from large geographical areas to local microhabitats (*Johnson, 1980*). Furthermore, *Matuszewski, Szafałowicz & Jarmusz (2013)* defined species indicators of cadaver relocation as those that at least (1) have a strong preference for a given geographical area or habitat, (2) are resistant to relocation disturbance, (3) live on cadavers, and (4) colonize cadavers shortly after death. Furthermore, common species are more likely to be found in association with criminal cases than are infrequent species. Unfortunately, an association with a specific habitat appears to be more pronounced in less common species than in more common ones (*MacLeod & Donnelly, 1957b*).

### Biogeography of European species of forensic importance

According to the common definition, the distribution of a species is the geographical area within which that species is observed. Species may not be uniformly distributed in this area: variation in local density (e.g., a clumped distribution) is common. However, individuals of a given species are not often observed outside of their distribution

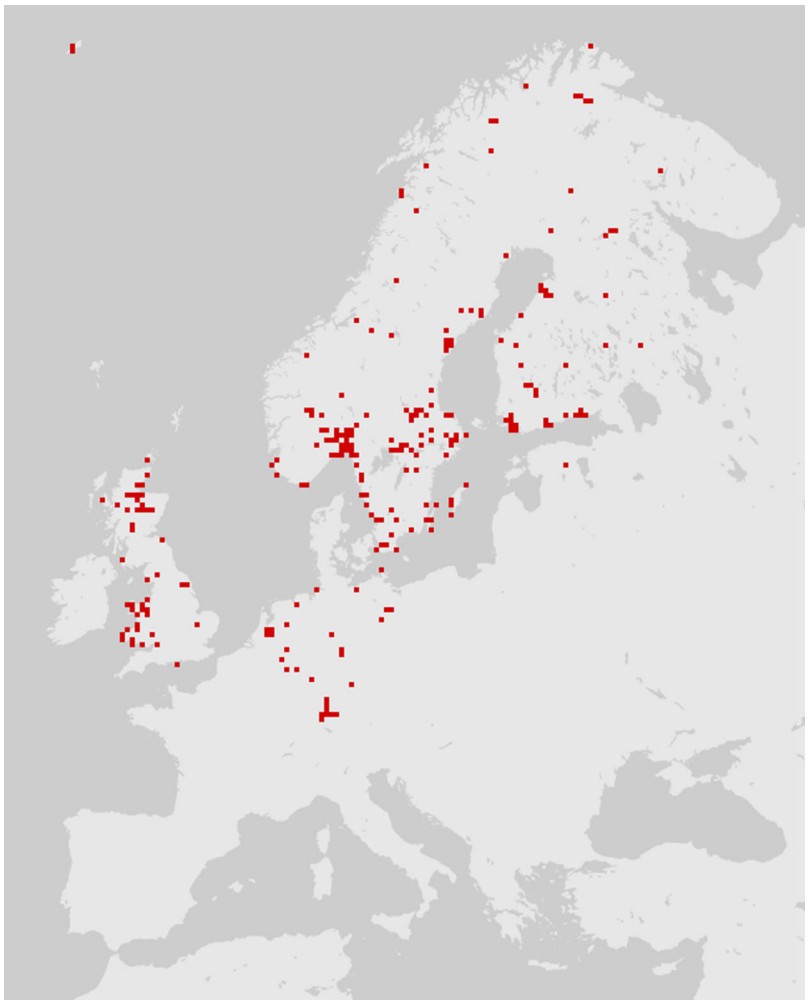

**Figure 1 The distribution of *Cynomya mortuorum* in Europe (source: http://www.gbif.org, 09/2016).** Although not reflected in the above map or represented in the source database, this species is present in northern France (*Bourel et al., 1999*), the mountains of central France and Italy (S. Vanin, 2017, unpublished data). This map is truncated due to a lack of published/registered data rather than to geographical restriction.

area. Online interactive maps can now be found on the web for most European taxa. Many of these databases combine older distribution data and contemporary records from amateur or professional entomologists (*Global Biodiversity Information Facility, 2017*). Such collaborative work is subject to information gaps and biases, particularly a lack of records; as a result, species distribution maps sometimes reflect the distribution of specialists more than the distribution of species (Fig. 1). In particular, a lack of records affects necrophagous species, which are infrequently sought out and are poorly known among entomologists. As a result, a necrophagous species detected in previously unsampled site may be considered unusual/unexpected while in fact being well within the species' distribution.

To be informative in determining cadaver relocation, a necrophagous species must have a restricted and well-established distribution. Herein, we list here the few European necrophagous species meeting these criteria.

Two common species of the genus *Cynomya* have restricted distributions within Europe. *Cynomya mortuorum*, a large, hairy bluebottle fly, can be found across the entire Palearctic region (*Braet et al., 2015*) (Fig. 1). But this species is rarely reported in central European countries, especially in a forensic context (*Rognes, 1991*; *Starkeby, 2001*; *Dekeirsschieter et al., 2013*; *Wyss, Cherix & Mangin, 2013*; *Braet et al., 2015*). Its distribution partially overlaps that of *Cynomya cadaverina* (Robineau-Desvoidy, 1830), another cold-adapted species of forensic interest (*Rognes, 1991*; *Kurahashi & Kuranishi, 2000*).

Two other calliphorid flies, *Calliphora loewi* and *Calliphora subalpina,* show a sub-alpine distribution (*Faucherre, Cherix & Wyss, 1999*; *Rognes, 1991*). In Europe, *Calliphora loewi* is mostly found in northern and central Europe, from Siberia and the Caucasus to the Central European Territories (*Smith, 1986*). Although this northern distribution area suggests that this species might be a good indicator of relocation, its recent discovery in Madeira Island (Portugal) calls its relevance into question (*Prado e Castro et al., 2016*). Furthermore, although it has a large distribution, *Calliphora loewi* is often recorded at low abundance, making sampling of this species on a relocated corpse unlikely (*Szpila et al., 2014*). *Calliphora subalpina* has a very similar distribution area and is subject to similar limitations (*Rognes, 1991*).

*Chrysomya albiceps* is one of the few species that is theoretically usable as an indicator of cadaver relocation in Europe (*Grassberger, Friedrich & Reiter, 2003*). The species is meridional, common and abundant in southern Europe and in most of the Neotropical, Afrotropical, and Oriental regions (*Grassberger, Friedrich & Reiter, 2003*). However, while it is mostly found in southern Europe, *Chrysomya albiceps* has been observed migrating northward during the hot summer months (*Wyss, Cherix & Mangin, 2013*). This northward expansion of its range has the potential to cause confusion and precludes its use as evidence of cadaver relocation. Similar factors affect the use of *Chrysomya megacephala*, an Asian fly recently recorded in continental Europe and extending its distribution into the Mediterranean region (*Martinez-Sanchez, Marcos-Garcia & Rojo, 2001*; *Ebejer, 2007*; *Prado e Castro & García, 2010*; *Bao & Wells, 2014*).

Some additional necrophagous species also have restricted distribution areas, but most of these species are unusual, difficult to identify and poorly documented. Thus, although they may be informative, these insects cannot be regarded as true indicators of cadaver relocation according to the criteria listed by *Matuszewski, Szafałowicz & Jarmusz (2013)*. According to this review, to date, insect distribution area as an indicator of long-distance cadaver relocation appears to be limited in application, with little utility for empirical forensics. However, the distribution of *Cynomya mortuorum* (Fig. 1) is a great example of an incomplete dataset that could be easily enhanced by forensic case databases and natural history collections targeted on carrion feeding insects. This example also highlights the potential of species distribution models based on environmental niche modeling, e.g., maximum entropy algorithm (MaxEnt) (*Szyniszewska & Tatem, 2014*; *Ali Hanafi-Bojd et al., 2015*; *Kumar, Yee & Neven, 2016*). Such models can provide probabilities of the locations of certain insects in a given area and could be used in a forensic context to determine the likelihood of cadaver relocation.
## Species-specific habitats

Many forensic cases involve cadavers that have been transported some kilometers from the crime scene, especially to low-traffic areas such as forests, dumping sites, rivers, or seashores (secondary decomposition sites) (*Matuszewski, Szafałowicz & Jarmusz, 2013*). As discussed above, such short-distance relocation cannot be elucidated using the presence of foreign necrophagous species. However, the transfer of a cadaver to a new location can alter micro-environmental conditions such as climate, extent of synanthropy, vegetation conditions, and indoor/outdoor location. The population of necrophagous insects at the secondary decomposition site may therefore differ from that of the initial (primary) environment.

The effect of habitat on the abundance of certain species is well established (*Hwang & Turner, 2005*; *Brundage, Bros & Honda, 2011*). However, published data regarding species-specific habitats vary, highlighting that such preferences are not fixed and often vary locally. The biology of necrophagous species contributes to this variability. To successfully reproduce, adult females must find suitable carrion for their offspring. However, the occurrence of cadavers is by definition unpredictable because death is temporally and spatially random. Accordingly, all necrophagous species have an efficient olfactory sense that allows individuals to quickly detect and locate cadavers. As noted by *MacLeod & Donnelly (1957b)*, blow flies are powerful and active flies capable of dispersing over large distances (several kilometers per day).

Furthermore, most studies in forensic entomology use simple descriptors (e.g., forest, sunny, and indoor) without taking into account the surroundings or variability within categories (e.g., forest type or city size). Additionally, larger-scale effects and interactions of parameters (e.g., temperature and city size) are typically not considered (*Zabala, Díaz & Saloña-Bordas, 2014*). In a 1957 study, MacLeod and Donnelly stated that "there is nothing to indicate whether the non-uniform distribution of the adult (flies) population is due to the faunal, floral, vegetation-structural or edaphic element of the environment, or to some combination of these." More than fifty years later, *Zabala, Díaz & Saloña-Bordas (2014)* concluded that, except for the summer abundance of *Calliphora vomitoria,* blow fly community composition could not be used as evidence of cadaver relocation, particularly in heterogeneous and densely populated areas. These authors also noted that conclusions based on species-specific habitat preferences should be drawn only on the basis of local studies (*Davies, 1990*; *Anderson & Van Laerhoven, 1996*; *Hwang & Turner, 2005*; *Brundage, Bros & Honda, 2011*). The following sub-section focuses on specific habitat characteristics that may be of relevance in determining the primary deposition site of a cadaver.

### Indoors vs. outdoors

The question of the inside/outside location of a cadaver is a key point in many forensic investigations (*Frost et al., 2010*). The location of a cadaver affects its colonization time (the pre-appearance interval, i.e., the time before insects reach the cadaver) and thus the post-mortem interval estimation (*Pohjoismaki et al., 2010*; *Reibe & Madea, 2010*; *Charabidze, Hedouin & Gosset, 2015*). Furthermore, access to the cadaver by necrophagous

insects greatly affects its decomposition (*MacLeod & Donnelly, 1962*; *Campobasso, Di Vella & Introna, 2001*). An indoor location also protects the cadaver from rain and is often associated with higher temperatures, which can increase the development rate of the larvae.

The species associated with indoor locations have been investigated in many field studies and case reports. A pioneer study by Goff of 35 forensic entomology cases in Hawaii noted that more insect species were found indoors than outdoors (*Goff, 1991*). During winter experiments, *Centeno, Maldonado & Oliva (2002)* found two more species on carrions that were sheltered than on unsheltered ones. *Anderson (2011)* found the same species (except *Lucilia illustris*) on both inside and outside cadavers. In contrast, *Cainé et al. (2009)* found more fly species on outdoor cadavers than on indoor cadavers in Portugal, and *Reibe & Madea (2010)* similarly found greater species diversity in outdoor locations. In the Reibe and Madea experiment, piglet carcasses located indoors (first-floor room) were exclusively infested by *Calliphora vicina,* whereas a variety of blow fly species (*Lucilia sericata, Lucilia caesar, Lucilia illustris, Calliphora vicina,* and *Calliphora vomitoria*) were found on the outdoor (garden) piglet carcasses. *Cammack et al. (2016)* recently published similar data on the decomposition of concealed and exposed porcine remains in North Carolina. According to this study, concealment had a significant effect on the insect community, and colonization was delayed by 35–768 h depending on the degree of concealment.

The importance of cadaver location was also observed for larder beetles, which preferentially feed and breed on dry material (*Charabidze, Hedouin & Gosset, 2013*). While feeding larvae were more common in indoor forensic cases than in outdoor ones, no clear preference was observed among adults. The same authors also found an effect of cadaver location on the presence of *Necrodes littoralis* (*Charabidze et al., 2016*). Leclercq reported Silphidae species only from cadavers recovered from forest sites in Belgium (*Dekeirsschieter et al., 2013*) and *Dekeirsschieter et al. (2011)* did not identify any Silphidae species in cadavers found in urban Belgium. However, *Chauvet et al. (2008)* recorded the presence of *Nicrophorus* spp. on human cadavers discovered inside houses in France.

In accordance with these discrepancies, *Frost et al. (2010)* noted that although more species and specimens tend to be observed indoors than outdoors, this trend is not consistent. An extensive table summarizing the insect species reported from human remains found indoors can be found in their study (*Frost et al., 2010*). The authors note that "none of the(se) listed insect species can be considered as exclusively indoors." An example of the difficulty in formally associating the presence of a species to the inside/outside location is provided by *Krikken & Huijbregts (2001)*. From the numerous dead *Lucicia* adult flies (no species name was reported) observed in an upstairs room with closed windows, the authors concluded that the body had first been outdoors in a warm, sunny environment and was later relocated into the room. However, this conclusion was based on the supposed preference of *Lucilia* to "oviposit on high temperature surfaces," which the authors interpreted as meaning "outdoors."

In the future, mites may provide information regarding the indoor/outdoor location of a cadaver, but mite species are poorly known and are currently overlooked in forensic

entomology. For further information, see *Frost et al.'s (2010)* abovementioned review and the work of *Perotti et al. (2009)*.

### Open vs. forest and sunny vs. shaded places

The distinction between open and forest habitat is not always clear: vegetation cover can vary according to season, and the location of a cadaver within an open habitat is not always sunny (e.g., in valleys). In a field study published in 1957, MacLeod and Donnelly found that *Calliphora vomitoria* and *Lucilia ampullacea* were abundant in regions of dense vegetation (i.e. forest habitats), whereas *Lucilia illustris* and *Lucilia sericata* were more common in open conditions (heliophilic species). In general, *Lucilia sericata* is often found in bright sunlight (*Holdaway, 1933*), whereas *Lucilia caesar* is associated with shade (*Nuorteva, 1964*). However, despite evidence of the thermophilic character of some blow fly species, preferences vary among local populations (*Martinez-Sanchez, Rojo & Marcos-Garcia, 2000*; *Hwang & Turner, 2009*). For example, *Joy, Liette & Harrah (2006)* found the same species on sunlit and shaded pig carcasses in West Virginia, USA, and *Hwang & Turner (2009)* showed the ability of *Calliphora vicina* populations to locally adapt their thermal requirements to suit their environment.

*Lambiase & Camerini (2012)* reported on the distribution of *Chrysomya albiceps* in northern Italy. During the summer of 2007, two cadavers were discovered next to each other in a mountainous wooded area. The two corpses showed differences in decomposition, and *Chrysomya albiceps* was found only on one corpse, suggesting that the victims were murdered at different times and places. Experiments with baited traps subsequently evidenced the absence of *Chrysomya albiceps* at the final location. As this species had recently expanded into northern Italy, the authors hypothesized that it had not yet reached all suitable habitats, especially those in mountains. Accordingly, they concluded that one corpse was colonized by *Chrysomya albiceps* at a lower altitude before being relocated to the mountain. This example was only possible due to the ongoing expansion of *Chrysomya albiceps* in this area and highlights the need for local studies and the performance of *a posteriori* experiments over the study of general trends.

Regarding coleopterans, *Matuszewski, Szafałowicz & Jarmusz (2013)* investigated species that colonized cadavers in open vs. forest habitats. They concluded that the presence of *Dermestes frischi*, *Omosita colon*, and *Nitidula* spp. could be used as evidence of relocation from rural open to rural forest habitat. In contrast, only *Oiceoptoma thoracicum* was classified as an indicator of relocation in the opposite direction. This conclusion is similar to that of *Dekeirsschieter et al. (2011, 2013)*, who recorded seven Silphidae species in forest habitat (Belgium): all but *Oiceoptoma thoracica* were also sampled in agricultural biotope (open habitat).

### Rural vs. urban

The term "synanthropic" is used to characterize species that live near humans and benefit from them and the artificial habitats they create. Cities, and more specifically human activities, are often associated with the production of meat waste that can attract necrophagous insects. Furthermore, urban areas are characterized by different

landscapes: high-rise buildings, urban sprawl, extensive areas of concrete or asphalt surfaces and a variety of infrastructure. These features affect the climate and result in local warming (*Wilby, 2003*). As ambient temperature is of prime importance for insect activity and development, heat islands such as those observed in large cities can offer thermal refuges for several species.

Although it is present in both rural and urban habitats, *Calliphora vicina* tends to be found predominantly in shady and urban areas (*Erzinçlioğlu & Bunker, 1996*; *Horenstein et al., 2007*; *Greco, Brandmayr & Bonacci, 2014*). In contrast, *Calliphora vomitoria* is often described as a more rural species that avoids cities (*Nuorteva, 1963*; *Povolny, 1971*; *Smith, 1986*; *Hwang & Turner, 2005*, *2009*). *Calliphora loewi* and *Calliphora subalpina* are also known to avoid urban areas (*Nuorteva, 1963*; *Rognes, 1991*; *Vanin et al., 2011*). In an extensive study examining a 7,000 km$^2$ landscape in Spain, Zabala, Díaze & Saloña-Bordas (2014) found a significant relationship between summer abundance of *Calliphora vomitoria* and both distance to urban areas and degree of urbanization. This pattern was especially clear during the summer, when *Calliphora vomitoria* was significantly more abundant at points far from urban areas. However, for the nine other calliphorid flies they investigated (including *Calliphora vicina* and *Lucilia sericata*), no clear synanthropic relationship was found.

Several comparative studies on rural and urban blow fly populations have been performed in the UK (*MacLeod & Donnelly, 1957a*, *1957b*, *1960*, *1962*; *Schumann, 1990*; *Isiche, Hillerton & Nowell, 1992*; *Smith & Wall, 1997*; *Davies, 1999*). Using meat-baited bottle traps, *Hwang & Turner (2009)* described three groups of necrophagous flies corresponding to three habitat types. The urban habitat was characterized by *Calliphora vicina, Lucilia illustris,* and *Lucilia sericata*, whereas rural grasslands were inhabited by *Lucilia caesar,* and the rural woodlands were inhabited by *Calliphora vomitoria. Wyss, Cherix & Mangin (2013)* reported that in Switzerland, *Lucilia argyrostoma* was found in urban areas, whereas *Cynomya mortuorum* avoided urban areas. *de Souza & Von Zuben (2012*, *2016)* found significant differences in the extent of synanthropy among some Calliphoridae and Sarcophagidae flies in Brazil. In contrast, in southern Africa, *Parry, Mansell & Weldon (2016)* observed that the species assemblages present in human-disturbed areas were very similar to those recorded in natural habitats.

However, most, if not all, species of forensic interest show inconsistencies or exceptions in their habitat-association patterns. For example, many authors have found that *Lucilia sericata* is associated with urban habitats (*Nuorteva, 1963*; *Isiche, Hillerton & Nowell, 1992*; *Fisher, Wall & Ashworth, 1998*; *Hwang & Turner, 2005*). A study from Germany found that *Lucilia sericata* had the highest synanthropy index (SI) of all blow fly species under study (*Steinborn, 1981* in *Reibe & Madea, 2010*). Another German study reported *Lucilia sericata* and *Calliphora vicina* as the only blow fly species caught indoors (*Schumann, 1990*). Similarly, *Lucilia sericata* was classified by *Greco, Brandmayr & Bonacci (2014)* as the most synanthropic blow fly in Italy. However, *Lucilia sericata* has also been recorded in natural open habitats in Poland and in open pasture in England (*Smith & Wall, 1997*; *Davies, 1999*; *Matuszewski, Szafałowicz & Jarmusz, 2013*). In Italy, *Vanin et al. (2008)* noted that *Lucilia sericata* does not show a habitat preference in regions with urban sprawl in rural

areas and cannot be used to ascertain whether a cadaver has been moved. Regarding *Lucilia caesar*, *Greco, Brandmayr & Bonacci (2014)* observed a preference of this species for wild and rural habitats, a trend supported by some previous studies (*Baz et al., 2007*; *Hwang & Turner, 2009*; *Greco, Brandmayr & Bonacci, 2014*), but in conflict with the findings of *Fisher, Wall & Ashworth (1998)*. Thus, while the presence of necrophagous species reflects their ecological preferences, these insects are not sufficiently discretely partitioned between urban and rural areas to be useful in determining corpse relocation in a forensic context.

There are few data on necrophagous coleopterans, likely due to the under-representation of these insects in anthropized environments. Due to their large size and low agility in flight, many Coleoptera species of forensic interest appear to be poorly suited to urban conditions. *Dekeirsschieter et al. (2011)* recorded seven Silphidae species in a Belgian forest environment, six in agricultural biotopes and none in urban locations. According to these results, silphid beetles may be good indicators of cadaver relocation between rural and urban habitats (*Matuszewski et al., 2010*; *Bala, 2015*).

## Other specific locations

### Water

One of the most readily identifiable type of relocation is that from water to open air. In such a case, the presence of any aquatic invertebrate on the cadaver could be evidence of cadaver relocation. In contrast, the finding of the typical necrophagous species on an immersed cadaver can be more challenging to interpret. *Merritt & Wallace (2001)* described four sequential steps of the change in body position in water over time: (1) the body sinks to the bottom, (2) horizontal movement occurs at the bottom, (3) the body floats to the surface, and (4) surface drift occurs. A cadaver discovered in water during the initial steps is characterized by the absence of the typical necrophagous species (e.g., Calliphoridae species) and the presence of ubiquitous aquatic invertebrates (e.g., Chironomidae larvae, snails). During the first two steps of immersion, the cadaver is fully immersed, and the presence of any terrestrial larvae on the cadaver would indicate that the eggs were laid before immersion. This possibility is interesting because blow fly larvae can tolerate submersion in water and can remain alive for several hours (*Abdel-Shafy et al., 2009*; *Reigada et al., 2011*). However, the finding of the same species on a floating cadaver (steps 3 and 4) would yield less or no information, as many fly species can lay eggs on the emerged parts of a floating cadaver (*Tomberlin & Adler, 1998*; *Barrios & Wolff, 2011*).

The presence of Coleoptera would be more informative for interpreting cadaver relocation. The larvae of most Silphidae species live underneath cadavers and dig pupation chambers into the soil for nymphosis. Thus, the larvae of these species should not be observed on floating cadavers. Furthermore, large adults are less agile in flight than are flies and thus avoid landing on small surfaces surrounded by water. *Barrios & Wolff (2011)* did not observe any necrophagous Coleoptera species on pig cadavers placed in two freshwater ecosystems, even during the floating phases. However, *Tomberlin & Adler (1998)*

observed many small staphylinid beetles on rat carcasses in water and found single adults of the silphid beetle *Necrophila americana* and the dermestid beetle *Dermestes caninus*. As dermestid beetles typically colonize and feed on dry materials (*Charabidze et al., 2014*; *Rosenbaum, Devigne & Charabidzé, 2016*), their findings highlight the risk associated with drawing conclusions regarding cadaver relocation from general trends.

The relocation of a cadaver from a freshwater to marine environment (and the inverse) sometimes occurs, especially where floating cadavers are carried by tides. As most aquatic species are limited to a narrow salinity range, the presence of a given species outside of this range may be evidence of cadaver relocation. Detailed data on species associated with marine and freshwater environments can be found throughout the literature (*Sorg et al., 1996*; *Anderson & Hobischak, 2004*). Cadaver relocation can also occur within the same aquatic environment. In freshwater, the species distribution depends on the physico-chemical attributes of the water (oxygen level, pollutant levels, and turbidity), and in running freshwater, there is a succession of habitats and biotopes from source to estuary. For example, Ephemeroptera and Trichoptera larvae are found only in clean and well-oxygenated water, whereas Eristalidae larvae are found in water with a high organic load (*Mason, 2002*). Abundant literature exists on this topic, especially with respect to bio-indicators (*O'Brien et al., 2016*).

*Insects of buried/concealed cadavers*

A common method of cadaver concealment is burial, which greatly affects carrion decomposition and access by entomofauna (*Simmons et al., 2010*). Deep burial and/or protection of the body by a coffin limit but do not prevent post-mortem colonization: Experiments on buried pig carcasses and insect sampling during exhumations have shown the presence of many necrophagous species (*Payne, King & Beinhart, 1968*; *Smith, 1986*; *Van Laerhoven & Anderson, 1999*; *Bourel et al., 2004*; *Lefebvre & Gaudry, 2009*; *Gaudry, 2010*). Although no necrophagous species appear to be restricted to buried cadavers, their relative abundance and diversity often differ between buried and exposed cadavers (*Braet et al., 2015*). Thus, the absence of one or more expected species (e.g., calliphorid flies) and the presence of many concealment-related species (e.g., Phoridae) may indicate that a cadaver has been previously buried (*Huchet & Greenberg, 2010*; *Huchet, 2014*).

*Conicera tibialis* is one of the main species found on concealed cadavers; several authors have reported that this small fly occurs frequently and in large numbers (*Bourel et al., 2004*; *Martin-Vega, Gomez-Gomez & Baz, 2011*; *Merritt et al., 2007*). The regular occurrence of *C. tibialis* on buried cadavers reflects this species' behavior: females can burrow through the soil to a depth of 2 m to oviposit, and larvae can burrow even deeper (*Mozayani & Noziglia, 2011*). *Megaselia scalaris* is also often found on concealed cadavers. This fly is a warm-climate species, but it has been carried around the world by humans and has been associated with indoor forensic cases in temperate regions (*Disney, 2008*). However, these two species are present in other environments, including indoors and the open air (*Disney, 2012*), and their presence cannot be considered definitive evidence of burial. Interestingly, *Szpila, Voss & Pape (2010)* demonstrated the ability of *Phylloteles pictipennis* and *Eumacronychia persolla* (Diptera: Sarcophagidae) to reach deeply buried

animal remains and reproduce on this food source. As noted by the authors, both of these species develop exclusively on buried food resources, making them potential indicators of cadaver relocation.

In contrast, common blow flies and muscid flies have limited abilities to colonize buried resources, although *Muscina stabulans* and *Muscina prolapsa* have colonized remains buried up to 40 cm deep (*Gunn & Bird, 2011*). As noted by the authors, the presence of large numbers of larvae of a given species feeding on bodies buried deeper than indicated by their species-specific limitations may be an indication that the body had been exposed above ground for sufficient time for eggs to be laid. Indeed, larvae laid before burying are able to fully develop on cadavers that were subsequently buried (*Bachmann & Simmons, 2010*; *Gunn & Bird, 2011*). *Gunn & Bird (2011)* also showed the ability of wandering larvae that developed on a buried cadaver to reach the surface and pupate. According to this finding, the presence of pupae on the soil above the grave does not indicate that the cadaver was buried after the pupal stages emerged. *Balme et al. (2012)* also showed that flies successfully reached the surface following burial at 50 cm depth as pupae or post-feeding larvae.

*Mariani et al. (2014)* reported the use of an unusual biocenose as evidence of post-exhumation entomological contamination. Entomological investigation revealed the presence of numerous necrophagous insects as well as omnivorous and storage pests (Dermestidae, Nitidulidae, and Tenebrionidae beetles; Tineidae moths; and cockroaches) on exhumed remains. As none of these insects are able to burrow as adults or larvae, their presence is evidence of contamination during storage in the cemetery after exhumation. Other inferences can be drawn from the absence or presence of specific instars, as described in more detail in the section of this review focusing on larval behavior.

## Conclusions related to species-specific habitats

As described herein in detail, the associations of species to specific habitats can be used to infer cadaver relocation from one habitat to another (Table 1). However, whereas entomological evidence related to species-specific habitats may help support or reject hypotheses regarding cadaver relocation, inferences unsupported by local data are typically not appropriate. Currently, most published information is based on restricted datasets; future work should focus on obtaining large amounts of data at the local scale. For example, *a posteriori* analysis of forensic entomology case databases could provide substantial information on the biology of necrophagous species (e.g., species found on indoor vs. outdoor corpses, seasonal prevalence of specific species) (*Dekeirsschieter et al., 2013*; *Disney et al., 2014*; *Charabidze et al., 2014*, *2016*; *Syamsa et al., 2017*). Back-testing field experiments should always be performed to verify the local applicability of interpretations.

Non-necrophagous species can also provide evidence of cadaver relocation. As reported by *Goff (2011)*: "If a body is outdoors near or under vegetation, it is possible for insects associated with that vegetation to move onto the body, although typically not to feed or lay eggs." However, as these insects are not directly associated with the cadaver, it may be difficult to conclude that they were moved together with it. Furthermore, the

 

**Table 1  Summary of main necrophagous insects species/instars that may suggest cadaver relocation.**

| Final location | First location | | | | | | |
|---|---|---|---|---|---|---|---|
| | Indoor (closed) | Outdoor/Other Rural | Urban | Forest | Freshwater | Salted water | Buried/Concealed |
| **Indoor (closed)** | | Numerous large adults beetles and ground-living Silphids larvae | | | Aquatic species, especially larvae (low displacement abilities) | | X |
| Outdoor/Other — Rural | ? | | X | ? | | | Mainly Phoridae, no/few Calliphoridae |
| Outdoor/Other — Urban | | ? | | ? | | | Many Phoridae with no other terrestrial species |
| Outdoor/Other — Forest | | ? | X | | | | |
| Outdoor/Other — Freshwater (immersed) | Many Phoridae with no usual Calliphoridae species | All typical terrestrial necrophagous species. Ground-living Coleoptera larvae would be especially informative | | | | Salted water species | |
| Outdoor/Other — Saltwater (immersed) | | | | | Freshwater species | | |
| **Buried/Concealed** | X | Calliphoridae, Sarcophagidae, Muscidae | | | Any aquatic species | | |

**Notes:**

The first location is represented in the columns, and the second (final) site is represented in the rows. Temperature and the length of time also influence the presence and abundance of each taxon; for this table, we considered the cadaver remained a few weeks at each location. Additional details and explanation for each scenario are presented in the main text. X, Not possible; ?, Questionable (Unknown/Insufficient data).

probability of having a non-necrophagous species (1) moving onto a cadaver, (2) being moved with the cadaver, (3) being sampled and identified at the secondary site, and (4) being located outside its natural range is likely very low. We have found no report of such a case in the forensic literature.

## BEHAVIOR AND DEVELOPMENT OF INSTARS

Extensive knowledge of the behavior of necrophagous insects is often key to interpreting forensic entomology. Knowledge of when adults are attracted on cadavers, how they colonize them and how their larvae grow allows forensic entomologists to elucidate a post-mortem chronology. However, this component of forensic analysis is frequently overlooked. Indeed, only a few studies focusing on the behavior of necrophagous insects have been published and most available data are descriptive, consisting of field observations or trends rather than quantitative experiments (*Tomberlin et al., 2011*). These restrictions make it difficult to draw indisputable conclusions. However, although insect behavior is not always quantifiable, it can be used to construct hypotheses and guide investigations.

### Adult behavior: colonization and egg laying

Egg laying on a cadaver is affected by climatic conditions, species behavior and cadaver accessibility: in theory, these parameters could be used to infer displacement between places with different climactic conditions. *Krikken & Huijbregts (2001)* reported a neat but questionable example of this concept. Based on the unverified assumption that *Lucilia sericata* females "oviposit on high temperature surfaces," the authors concluded that a cadaver discovered in a room had first been located outdoors in a warm and sunny place. However, although *L. sericata* is heliophilic, it can lay eggs indoors.

Though a species-specific minimum temperature is required for egg laying (*Hédouin et al., 1996*; *Faucherre, Cherix & Wyss, 1999*; *Ody, Bulling & Barnes, 2017*), several other weather parameters, such as the levels of sun, wind and rain, can also have effects on egg laying. Fly displacement and egg laying primarily occur during the daytime (*Wooldridge, Scrase & Wall, 2007*; *Greenberg, 1990b*; *Benecke, 1998*; *Wells et al., 2001*). Accordingly, the presence of numerous egg batches on a cadaver located in a dark place suggest the possibility of cadaver relocation. However, *Faucherre, Cherix & Wyss (1999)* reported a case where *Calliphora vicina* females had flown to and oviposited on a cadaver in a 10-m-deep cave located in the Swiss Jura mountains, and *Gemmellaro (2016)* recently showed the ability of some calliphorid flies, especially *Calliphora vicina*, to reach meat-baited traps placed inside volcanic caves.

Some species are known to oviposit in specific areas of a cadaver. For example, calliphorid flies preferentially deposit egg batches on the face (nostrils, mouth, and eyes), whereas most silphid beetles lay their eggs underneath the cadaver (*Smith, 1986*). However, the region of oviposition is strongly affected by the extent of cadaver decomposition or wounds, the presence of other larvae and species, collective behavior (egg aggregation) and environmental characteristics (*Charabidze et al., 2015*). On the other hand, strong evidence of cadaver relocation is provided by the presence of eggs,

especially those of large calliphorids, in inaccessible places, such as underneath a cadaver. Such a case was analyzed in France (D. Charabidze, 2013, unpublished data): the presence of numerous *Lucilia sericata* and *Calliphora vicina* egg batches in the folds of clothing underneath a cadaver served as evidence of the secondary reversal of the cadaver by drug addicts searching for money.

## Larval development, wandering larvae, and pupae

Fly larvae live on the cadaver and are thus very resistant to cadaver relocation. In contrast, the probability of transferring insects present in the soil or below the cadaver (wandering larvae, pupae, and most silphid larvae) along with the cadaver is very low. Accordingly, considerable information can be obtained from the presence and location of wandering larvae and pupae/puparia around a cadaver.

During the post-feeding stage, the larvae of several blow fly species (with the notable exception of Chrysomyinae species) begin to migrate from the cadaver to pupate in a protected location away from predators (*Gomes, Godoy & Von Zuben, 2006*). *Greenberg (1990a)* observed that more than 80% of post-feeding larvae of *Lucilia sericata* and *Calliphora vicina* removed of the cadaver to a distance up to 8 m away. In contrast, only 2% of *Phormia regina*, 10% of *Muscina stabulans*, and 16% of *Chrysomya rufifacies* larvae moved away from the cadaver. However, *Lewis & Benbow (2011)* reported cases of *en masse* post-feeding dispersal of *Phormia regina* larvae away from swine carcasses in experiments conducted during the summer. This unexpected observation illustrate the need for careful case-by-case analysis before drawing conclusions.

The presence of necrophagous blow fly pupae and puparia (or dead adult flies) in an empty place can be evidence of the former presence of a cadaver. This question was extensively discussed during the famous Casey Anthony trial (USA) (*Lohr, 2011*), in which a first forensic entomologist relied on the presence of numerous *Megaselia scalaris* larvae, pupae and adults in a car trunk as evidence of the former presence of a cadaver. However, an expert witness for the defense showed that the same insects could have come from a trash bag discovered in the trunk. As the gut contents of insect samples were not tested for DNA, there was no evidence to support the assertion that the insects originated on human remains (*Campobasso et al., 2005*, see the molecular analysis section of this review). *Benecke (2004)* described a similar case. The corpse of a man was discovered in the trunk of his car. Only a few larvae and pupae were sampled on the cadaver, but no pupae was found in the gaps of the trunk. The author postulates that the cadaver was stored in another place until post-feeding larvae left the corpse. Afterward, the corpse was moved into the car trunk, explaining why only a few maggots were found on the cadaver. Conversely, *Krikken & Huijbregts (2001)* reported a skeleton found during the winter in a small forest with "numerous empty pupal cocoons of *Protophormia terraenovae* under the bones." Based on the presence of these puparia, the authors concluded that the entire decomposition process had taken place in that same place. However, as this species can pupate in clothes or on decomposing tissues, it is possible that the pupae had been moved together with the cadaver. Lastly, *Mariani et al. (2014)* observed that blow fly and muscid larvae buried with a cadaver ultimately left their food source to move to their typical

pupariation depths. According to those authors, the presence of large numbers of post-feeding blow fly larvae without a cadaver in the vicinity could indicate that a body may have been buried nearby rather than relocated.

Cadaver relocation can sometimes be characterized by discrepancies between local temperature and larval development. For example, the presence of third-instar *Lucilia sericata* individuals on a cadaver located in a cold location (e.g., a cellar with a constant 9–10 °C temperature) would suggest relocation. However, such discrepancies often result from changes in microclimate (e.g., direct sun exposure vs. shade), larval-mass effect or conservation of the samples (e.g., high temperature during transport) rather than from relocation. Cadaver relocation should be considered as a possibility only in the absence of these other influences.

Lastly, the presence of crushed pupae/imago on or under the cadaver can be indicative of relocation. We observed the presence of flattened pupae or newly hatched flies (flat, dry individuals with a still-visible ptilinum) directly under the cadaver in several forensic cases. In these cases, relocation of the cadaver had likely occurred after numerous flies had started to emerge, and some specimens were compressed under the cadaver during or after moving. If such specimens are observed on site (before the cadaver is moved by the forensic team), relocation after larval pupation might have occurred.

## PHENOLOGY AND COLONIZATION TIME

The temporal activities of insects vary due to intrinsic properties (e.g., life history, reproductive cycle, and development time) and extrinsic seasonal effects (e.g., temperature, photoperiod and resource availability) (*Zabala, Díaz & Saloña-Bordas, 2014*). Species-specific phenology can therefore be used as an indicator of the season of death and, at least theoretically, of cadaver relocation (*Joy, Liette & Harrah, 2006*). For example, the finding of only "late" colonizers on a cadaver with no traces of pioneer species suggests that the cadaver was not accessible to insects during the first stages of decomposition. An example is given by *Krikken & Huijbregts (2001)*: only a small number of insect eggs (attributed to blow flies) were found on a cadaver discovered during a warm summer. Considering the total absence of maggots from the body and the post-mortem interval calculated by the pathologist, the authors concluded that the body must have been sheltered, delaying colonization by blow flies.

*Madra, Konwerski & Matuszewski (2014)* observed clear seasonality trends for nine Staphylininae species and concluded that these species are good candidates as indicators of cadaver relocation. The results for flies are more divergent, and the interactions of phenology and spatial distribution prevent the use of most species as indicators of post-mortem displacement. In the USA, *Cammack et al. (2016)* reported a significant seasonal effect on the colonization of piglet carrion by blow flies: *Lucilia illustris* (Meigen) was indicative of spring colonization, *Cochliomyia macellaria* (F.) and *Chrysomya megacephala* (F.) were indicative of summer colonization, and *Calliphora vicina* and *Calliphora vomitoria* were indicative of fall colonization. In Spain, *Zabala, Díaz & Saloña-Bordas (2014)* observed that *Lucilia sericata, Lucilia illustris,* and *Chrysomya albiceps* were clear indicators of summer colonization, whereas *Calliphora vicina* and *Calliphora vomitoria*

were common year round, with maximum abundance in the spring. However, due to wide variation with landscape, the authors concluded that these species should not be used as indicators of cadaver relocation. The results from a study in Italy led to a similar conclusion (*Greco, Brandmayr & Bonacci, 2014*). Although the authors detected differences in the abundances of Calliphoridae taxa among sampling months, the effect was strongly dependent on trap location. For example, *Calliphora vicina* was observed throughout the sampling period (except from June to September) in rural and urban areas but was absent during the cooler months (November to January) in the wild area. Similarly, *Vanin et al. (2008)* reviewed several Italian forensic cases and concluded that further studies are necessary to confirm whether *Lucilia sericata* can be used to estimate the season of death.

Evidence of relocation can also be inferred based on the stage of decomposition at which a necrophagous species colonizes a cadaver. This subject is widely studied and debated within the forensic entomology community. It is well known that some species are early colonizers, whereas others are observed during later stages of decomposition (*Smith, 1986*). However, the colonization period of a given species varies depending on many parameters, including climate, season, geographic area, local environment, and insect population (*Campobasso, Di Vella & Introna, 2001*). These influencing factors must be carefully examined before unusual succession can be considered as evidence of cadaver relocation. Furthermore, open habitats allow easy access to the cadaver for predators or parasites, such as wasps, Silphidae and Cleridae beetles. They can decrease the number and diversity of Diptera larvae, especially if predation occurs during the early developmental stages (e.g., egg removal by wasps). Thus, the absence of some pioneer species does not necessarily indicate cadaver concealment during the first stage of decomposition.

Finally, different amounts of time spent in the first location, during transportation and in the secondary decomposition site are associated with different types of evidence. Table 2 summarizes the overall scenarios and corresponding timeframes for "simple" cases. However, the problem of time must be considered for each particular case.

# CONTRIBUTION OF MOLECULAR ANALYSES

## Cuticular hydrocarbons

The ability to identify forensic species at different developmental stages and their associations with local populations can be crucial in determining whether a body was moved from a crime scene. Simple molecular analyses concern cuticular hydrocarbon profiles. Insects form a thin, epicuticular layer of wax consisting of free lipids, which is a class of compounds that includes hydrocarbons, alcohols, fatty acids, waxes, acylglycerides, phospholipids, and glycolipids (*Gibbs & Elizabeth, 1998*). The cuticular hydrocarbon phenotype is biologically very stable and almost entirely determined by genotype (*Wells & Stevens, 2010*; *Pechal et al., 2014*). *Byrne et al. (1995)* showed that the cuticular hydrocarbons of three geographically distinct populations of *Phormia regina* are differentiable. However, some local populations can interbreed with adjacent populations, and the minimal interval over which adjacent populations can be considered distinct remains unknown. Accordingly, this method could be used to identify the presence of a

**Table 2 Effects of time spent in the first (columns) and secondary (lines) decomposition sites on the presence/stage of necrophagous entomofauna.**

| Duration at the second deposition site | Duration at the first deposition site | | | |
|---|---|---|---|---|
| | Hours | Days | Weeks | Months/years |
| Hours | None (duration is insufficient to sample insects from the 1st location) | Only Calliphoridae larvae from the first site | Various species from the first site only | |
| Days | | Calliphoridae larvae from both sites | Various species from the first site + late colonizers from the second site | Predominantly late colonizers from the first site |
| Weeks | | Empty pupae (non-wandering fly species) from the first site | | Late colonizers from both sites |
| Months/years | | | Traces of various species from the first site + late colonizers from the second site | |

Note:
 Species and developmental instars on the cadaver can vary with the time spent in each location, affecting the interpretation of entomological samples as evidence of cadaver relocation. Additional details on the entomological phases of the colonization process can be found in *Smith (1986)*.

non-local population on a cadaver (which suggests cadaver relocation) but will not be informative in cases of short-distance relocation (*Byrne et al., 1995*). More research on this promising topic should be conducted in the future (*Tomberlin et al., 2011*).

## Genetics of insect populations

DNA-based identification is valuable for identifying adult insects of forensic interest as well as immature stages, cuticle fragments, and puparia (*Sperling, Anderson & Hickey, 1994*; *Zehner et al., 2004*; *Guo et al., 2010*; *Li et al., 2010*; *Wells & Sperling, 2001*; *Mazzanti et al., 2010*; *Xinghua et al. 2010*; *Guo et al., 2011*; *Zaidi et al., 2011*; *Jordaens et al., 2013*; *Meiklejohn, Wallman & Dowton, 2013*; *Park et al., 2013*, *Sonet et al., 2013*). If post-mortem changes are suspected, relocation can be evidenced by determining the relationships between insects sampled at the initial and secondary sites (*Picard, Villet & Wells, 2012*). Several studies have highlighted significant genetic differences between populations of the same species within (*Hall et al., 2001*; *Boehme et al., 2013*; *Desmyter & Gosselin, 2009*; *Sonet et al., 2013*) and among continents (*Hall et al., 2001*; *Lyra et al., 2005*). To identify genetic variation among populations, methods such as simple conformation polymorphism strand analysis (SSCP) and amplified fragment length polymorphism (AFLP) are available. However, kinship analyses require a solid genetic database that encompasses variability among geographical sites, the development of which requires thorough field sampling, precise morphological identification and the complete genetic characterization of each collected individual.

Databases such as GenBank and the *German Barcode of Life (2017)* have been used to detect genetic variation within local populations of the same species (*Wells & Stevens, 2010*). Using SSCP analysis, inter-population differences have been detected between African and North American populations of the common housefly, *Musca domestica* (*Marquez & Krafsur, 2003*). *Harvey et al. (2003)* found differences in the COI gene between South African and Australian populations of two species of forensic interest,

*Chrysomya rufifacies* and *Lucilia cuprina*. Furthermore, *Desmyter & Gosselin (2009)* and *Boehme, Amendt & Zehner (2012)* found sequence differences between *Phormia regina* specimens from North America and Europe. *Jordaens et al. (2013)* confirmed this divergence in the COI gene between *P. regina* populations with newly sequenced material. However, sequence divergence within each continent was only ca. 0.4%, making genetic differentiation of local strains difficult to detect. At the local scale, blow flies show genetic isolation by time but not distance. Scientific projects dedicated to building datasets that reflect the diversity of necrophagous entomofauna at the European scale are currently expanding and are expected to address this topic in the near future (*Sonet et al., 2013*; *Geiger et al., 2016*).

Using AFLP surveys, Picard and Wells observed that groups of adult *Lucilia sericata* and *Phormia regina* individuals trapped together on a bait were predominantly composed of related individuals, with a level of genetic diversity lower than that observed at a larger scale (*Picard & Wells, 2009*, *2010*). This pattern is also observed in gravid females and is likely exhibited by larvae, suggesting that the population genetic structure of adults could be extended to the larval population growing on a cadaver. Such a finding would support the potential use of genetic tests to infer post-mortem relocation of a cadaver: for example, a larva found in one location might belong to a larval population growing on a cadaver in a second location. *Faulds, Wells & Picard (2014)* confirmed the validity of the AFLP method, finding that kinship testing based on AFLP data yielded adequate kinship estimates with limited error. As noted by those authors, this type of analysis can be performed on any life stage of the insect and on any species. Regarding species of interest in forensic entomology, AFLP data are currently available for *Phormia regina*, *Lucilia sericata*, and *Chrysomya megacephala* (*Picard & Wells, 2009*, *2010*; *Bao & Wells, 2014*). AFLP analysis of full sibship is a promising method for the detection of post-mortem relocation.

Many interesting ideas and examples can be found in the literature on wildlife genetic geographic origin assignment. DNA testing, which relies on the assignment of an unknown sample to its genetic population of origin, has become a widespread tool in wildlife forensic science. The excellent review by *Ogden & Linacre (2015)* provides not only examples and detailed methods of DNA testing, but also valuable consideration on its use in court. As noted in the review, the availability and quality of reference data are of paramount importance and are currently the main disincentive of the application of DNA-based methods in the field of forensic entomology.

## Identification of human DNA

Another contribution of molecular analysis is the identification of human DNA in the digestive tract of the larvae. This method can be used to determine the genetic profile of the victim (*Campobasso, Di Vella & Introna, 2001*; *Wells & Sperling, 2001*; *Benecke & Wells, 2002*) and can be used as evidence in the absence of a cadaver (*Gaudry et al., 2007*). Indeed, the presence of necrophagous larvae or pupae in an otherwise barren location can suggest the former presence of a cadaver. If genetic analysis of the gut contents of larvae reveals the victim's DNA, entomological evidence can be used to infer

relocation (*Wells et al., 2001*; *Lohr, 2011*). In 2001, Wells et al. showed that mitochondrial DNA sequences can be obtained from the dissected gut of a maggot that had fed on human tissue. In *Chavez-Briones et al. (2013)* reported the first forensic case of victim identification from human DNA isolated from the gastrointestinal tract of necrophagous larvae. *Njau et al. (2016)* showed that DNA analysis could be used to determine whether the larvae sampled on a cadaver were introduced from an alternative food source (e.g., a dead animal or a trash can near the cadaver). However, due to the rapid degradation of DNA by gut digestive enzymes, such analyses are limited to two days post-feeding (*Picard & Wells, 2009*; *Charabidze, Hedouin & Gosset, 2013*). Such restriction may change in a near future: a recent paper by *Pilli et al. (2016)* successfully used next generation sequencing to obtain a human profile from the gastrointestinal tract of head lice. Additionally more striking evidence of the potential of this method was provided by *Marchetti et al. (2013)* and *Madra, Konwerski & Matuszewski (2014)*. They used short tandem repeat (STR) analysis to extract and type human DNA from empty puparia collected in two forensic cases. As puparia cases are highly durable, they offer a unique opportunity to indicate cadaver relocation a long time after the event.

## CONCLUSION

1. The issue of cadaver relocation has arisen in many forensic cases and has received particular attention in forensic entomology.

2. Although some species are preferentially found in specific biotopes, most are not sufficiently geographically restricted to serve as indicators of cadaver relocation.

3. Time is a key factor influencing the presence of necrophagous insects. A cadaver that remains only briefly in the first location is unlikely to be colonized by local insects, whereas a cadaver that remains for a very long period will have been abandoned by insects before cadaver relocation occurs.

4. Circumstances that allow the clear inference of corpse relocation based on cadaver entomofauna are the following:

   – Relocation from open air to an aquatic environment (and vice versa),
   – Relocation from open air to a grave or burial site (and vice versa),
   – Removal from an indoor location at which some larvae or pupae remain, and
   – Genetic associations between larval populations or identification of human DNA.

5. Only field studies performed at the local scale and focused on a clear question (e.g. differences between indoor and outdoor entomofauna) should be used as references.

6. We recommend that forensic entomologists perform local trapping and experiments *a posteriori* to comply with the circumstances of a given forensic case. Corpse relocation inferences should not be based on general trends or previous results at a broader scale.

7. Analyses should be performed only by trained forensic entomologists and require early discussion with investigators, extensive on-site sampling, the conservation and analysis of relevant samples, and a considerable amount of chance.

8. Future work should focus on sharing and analyzing forensic entomology case databases. Such studies are less time-consuming than field experiments and can reflect a variety of circumstances; thus, they have the potential to provide abundant information on the biology of necrophagous species of forensic interest.

### Funding
The authors received no funding for this work.

### Competing Interests
The authors declare that they have no competing interests.

### Author Contributions
- Damien Charabidze analyzed the data, wrote the paper, prepared figures and/or tables, and reviewed drafts of the paper.
- Matthias Gosselin analyzed the data and reviewed drafts of the paper.
- Valéry Hedouin reviewed drafts of the paper.

### Data Availability
The research in this article did not generate any data or code (literature review).

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
