# Peer review of "Use of necrophagous insects as evidence of cadaver relocation: myth or reality?"

_PeerJ, doi:10.7717/peerj.3506_

## Round 0.1 · original submission · Minor Revisions

The review is very interesting, but contains a small number of easily correctable errors (e.g. grammar, missing references). I recommend to follow the reviewers' suggestions prior to acceptance.

Reviewer 1 ·

Basic reporting

A few minor edits for language required.

blow fly is spelled as two words per standard entomological rules.

analysze is a misspelling

line 111 "appears being a complex" is an awkward statement

line 153-154 is confusing

line 445: Is there supposed to be a species name after L.?

line 513: Focus on flies necessary?

A few additional citations or conversations with colleagues may help mature some concepts.

Indoor/outdoor: See Michelle Sanford's BioRxiv prepublication on indoor/outdoor occurrences in Houston, TX.

For Open Forest Sunny Shaded: Talk to Sherah VanLaerhoven. She gave a good ecology based talk on this topic for blow flies in wooded fields versus corn fields at the International Congress of Entomology.

Indicator species: Jonathan Cammack did some work in South Carolina and identified indicators of a few points of interest noted in this document.

Phormia larval movement: See work by Eric Benbow on larval mass movement.

Statement ending at line 185 may not be true for US, Russia, China, Australia, etc. Especially in the US, there is considerable movement across state lines and among regions of a continent.

There is a tendency for the authors to make general logical statements (is unlikely), which appear to be as unfounded as the statements they critique. I suggest rephrasing such sentences (is unsupported by current knowledge?).

On the genetics: blow flies show genetic isolation in time, but not by distance. For some applications, genetic isolation by distance would be more informative here, suggesting a need for similar projects with other types of organisms. Also, look at the forensic wildlife literature. There are great "population" assignment examples, with casework relevance, in that community. Also, this community negates the author's point that only entomologists consider postmortem movement. As genomes and transcriptomes for carrion insects become available, these applications will become more and more feasible.

Experimental design

This is a generally well written review with a great deal of effort and attention given to the topic.

Validity of the findings

I generally agree with most of the document, though some statements about probabilities of events are just as unfounded (as currently written) as the statements they are meant to critique.

Probabilities of the locations of certain insects could be provided and speak to the increased or decreased likelihood of movement.

One major point that seems apparent while reading the document is the need for more general research on carrion feeding insects, as noted in Tomberlin et al. 2011 (TREE not ARE). Specifically, the example of C. mortuorum is a great example of an incomplete data set that could be used to develop a maximum entropy (or similar species distribution) model for the expected distribution of a species. Such efforts would be greatly enhanced by natural history collections targeted on carrion feeding insects. Ideally, carrion insects with restricted ranges would be most informative, but any species found outside its expected range could be associated with a probability statement associated with species distribution models. This example highlights the need for basic collections development and species distribution modeling. Readings in wildlife forensics will reinforce these concepts.

Additional comments

Overall this is an excellent review. One frustrating aspect of the document is that it comes off as presenting what is wrong with current thinking, without much effort to show how the field can move forward explicitly. For instance, areas of work are noted, but it is not clear if they should be research priorities or if they are just held up as historical examples with limitations. Clear examples of research priorities (and a rationale for those specific goals) that are expected to move the community forward would help immensely. One example: should all casework reviews provide probabilities of indoor/outdoor species occurrences (as seen in Sanford Rxiv paper)? What are the types of reporting/research that the authors feel will advance the field most effectively in this area?

Reviewer 2 ·

Basic reporting

The article is clear text and it is conformed to professional standards of courtesy and expression. The topic is very interesting and concerns a real problem of some crime investigations.
It includes a clear introduction and the chapter "References" is rich also if it lacks of some references that could be of interest
The structure of the article has a general good format, excepted for the list of contents: incoherence in the succession of the letters that name the chapters and those of the inner sections; incoherence with the titles of the chapters into the text.
The only figure has sufficient resolution, and is appropriately described but it represents only an example, of course, and maybe it is not so relevant: the text explains the problem clearly.
The two tables are not completely intuitive, at the first visual impact, and have different shape. In the Table 1, the first row "Duration on first deposition site" probably might cover all the four columns below (hours, days, weeks, .....). A regular rectangular shape, as in the Table 2, could be nicer.
The article represent an appropriate ‘unit of publication’, and includes several suggestions relevant to the topic.

Experimental design

The article defines the problem quite clearly and allows to well detail the numerous theoretical approaches to the topic through the collected references.
Being the article a review, the only described method is related to the finding of the literature and its analysis, but the aim is clear and focuses a frequent problem in applied forensic entomology.

Validity of the findings

The findings are represented by the classification and the organization of the data extracted by the references.
The conclusion are well stated, but in the case of C. albiceps, for example, it is described as not useful species,because of its larger distribution. However this species is confined in particular habitat and in some cases it has been the only species to confirm the cadaver dislocation. One of these cases is described in “Lambiase S., Camerini G. 2012. Spread and Habitat Selection of Chrysomya albiceps (Wiedemann) (Diptera Calliphoridae) in Northern Italy: Forensic Implications. Journal of Forensic Sciences. 57:799-801” one of the most recent paper in which the removal of cadaver has been established on the base of the collected entomological evidences, not congruent with the recovery environment.

Additional comments

Reading this paper has been stimulant and confirm the needs to carry on different types of research to improve the applied use of forensic entomology.
Particular enphasis must be given to "a posteriori experiments" (conclusion, point 6) to ensure the best comprehension of a forensic case and to enrich the knowledge.

Reviewer 3 ·

Basic reporting

The articles need a language revision, some part need to be re-written in english

More attention to the references needed

The ms that is a review is interesting and properly summarized the main point of the "cadaver relocation" topic. Additional observations have to be added to the text as reported on the pdf

Tables are usefull and well explained

Experimental design

This is not an experimental work, but a review.
Lack of information is reported on the text

Validity of the findings

This is a speculative work based on a review, it is clearly stated since the begining of the work

Additional comments

Interesting work, well structured but it need some adjustments first of all the english revision

---

## Round 0.2 · accepted · Accept

I think that now the manuscript is ready to be published. Congratulations!